# 3D Printed Microfluidic Bioreactors Used for the Preferential Growth of Bacterial Biofilms through Dielectrophoresis

**DOI:** 10.3390/mi13091377

**Published:** 2022-08-24

**Authors:** Alexandra Csapai, Dan A. Toc, Florin Popa, Nicoleta Tosa, Violeta Pascalau, Carmen Costache, Alexandru Botan, Catalin O. Popa

**Affiliations:** 1Materials Engineering Department, Technical University of Cluj-Napoca, 103-105 Muncii Ave., 400641 Cluj-Napoca, Romania; 2Department of Microbiology, Iuliu Hatieganu University of Medicine and Pharmacy, 8 Victor Babeș Street, 400000 Cluj-Napoca, Romania; 3National Institute for Research and Development of Isotopic and Molecular Technologies, Molecular and Biomolecular Physics Department, 67-103 Donat Street, 400293 Cluj-Napoca, Romania

**Keywords:** bioreactors, microfluidics, dielectrophoresis, additive manufacturing, biofilms

## Abstract

A realistic modelling of the way biofilms form and evolve in time requests a dynamic approach. In this study, the proposed route uses continuous-flow bioreactors under controlled flow rates and temperature in the culture medium containing bacteria or fungi. 3D printed, Polylactic acid (PLA), flow-based bioreactors with integrated copper electrodes were used to investigate the effect of dielectrophoresis on the formation and growth of Staphylococcus aureus ATCC 25923, Enterococcus faecalis ATCC 29212, Pseudomonas aeruginosa ATCC 27853, and Klebsiella pneumoniae ATCC 13883 biofilms. Bacterial suspensions of 1McF turbidity have been prepared and circulated through the bioreactors. At the same time, a 30 V potential difference was applied on the system. The effect of the non-uniform electric field induced upon the bacterial cells was determined using quantitative methods, such as an adjusted microtiter plate technique, as well as spectral domain optical coherence tomography (SD-OCT) images. The morphology and the surface quality of the biofilms were investigated using Scanning Electron Microscopy (SEM) images. The results show that the different bacterial cells present a positive dielectrophoretic behaviour, with the preferential formation of biofilms in the high field gradient region.

## 1. Introduction

Although many efforts were made towards developing the molecular biology and genetic manipulation technologies in the last quarter of the century, the need to improve productivity, to facilitate screening of recently discovered microorganisms, to optimize the process, and to improve it remains. Small scale bioreactors answer to this necessity by providing miniaturized high throughput (HT) solutions to process development [1].

The most widely used investigation techniques of cell interactions with the surrounding environment include the use of large-scale bioreactors, with working volumes of several hundreds of cubic meters. However, these large-scale bioreactors present a substantial disadvantage represented by the inhomogeneous conditions provided inside, caused by imperfect mixing. Exposing cells and microorganisms to fluctuating pH, temperature, nutrients, and dissolved oxygen can have detrimental effects on the cell physiology and therefore on the overall process performance. In contrast, single-cell analysis techniques enable the investigation of individual cellular behaviour, allowing accurate control over environmental conditions, as well as the use of small quantities due to increased volume to surface ratio [2]. By overcoming the limitations of current available technologies, microfluidics has the capacity to enable the development of systems biology. Several advantages, such as the small-scale, sample quantities of nanolitres and the ability to increase the speed of common experimental procedures by orders of magnitudes advocate for the use of microfluidics for applications such as biomedical applications [3,4] high throughput drug screening [5,6], biological and chemical sensing [7,8], and genetic analysis [9]. However, classical microfluidic technologies also present some limitations, such as the necessity for special equipment for manufacturing and running systems. Due to the fabrication process they can become expensive, and the operations around the systems can become tedious, with frequent clogging problems emerging in the channels [10,11,12,13].

An emerging technology that could facilitate time and cost reductions in the process of microfluidic systems production is Additive Manufacturing (AM) or 3D printing. AM allows the production or reproduction of complex structures or shapes in two steps: conversion of a deigned 3D model into printed structures and placement of the printer head in different points along the three directions in space to print the object layer-by-layer [14].

The last few years brought attention towards FDM, the most used AM technology, not just as a technology for prototyping but as a valid means of production for final parts. The FDM process involves the use of a thermoplastic material, heated up to a semi-liquid stated, pushed out through an extruder, and deposited layer-by-layer on a print bed [15]. The obtained parts can be used as they are or can be further processed to improve their surface quality. One of the most frequently used materials in the FDM process is the PLA, a biocompatible, biodegradable thermoplastic aliphatic polyester, obtained from non-fossil renewable natural sources through the process of fermentation of polysaccharides or sugar [14]. One concern regarding FDM that may emerge is the lack of sealing between the deposited layers, as opposed to the Stereolithography (SLA) approach, which often performs better in such applications. However, both the bath (object is submersed into the resin and polymerized by a UV beam) and the constrained surface (object is pulled out of the resin, rather than submersed in it) SLA techniques present a substantial disadvantage: the object is printed hanging from the movable substrate [16]. This method leads to gravitational forces acting upon the resin droplets, which later can lead to the clogging of the channels (both central and inlet/outlet channels). Therefore, for this research, an FDM fabrication method was decided upon.

A critical factor for microfluidic-based cell analysis, sorting, isolation, or encapsulation is the capacity to manipulate microscale bioparticles. Various manipulation mechanisms can be employed based either on extrinsic or intrinsic properties of the bioparticles [17]. A common intrinsic-property-based cell manipulation technique for cell-trapping and selective cultivation is dielectrophoresis (DEP) [18,19]. DEP forces are described as the forces induced upon a polarizable particle suspended in a non-uniform electric field. Based on the response of the particle to the non-uniform electric field, whether it is attracted towards high field gradient region (positive DEP) or pushed away from it (negative DEP), its trajectory can be predicted, and therefore, DEP forces can be used to selectively and preferentially manipulate cells [1,20]. The main advantages in using DEP for bacterial cell manipulation are the possibility of label-free manipulation with high efficiency, biocompatibility, sensitivity, and controllability [17]. An application of the DEP in a microfluidic bioreactor could be the preferential formation and growth of biofilms. Scientific literature describes biofilms as communities composed of bacterial and/or eukaryotic cells encased in an extracellular polymeric substance, produced as a tri-dimensional structure by the microorganisms [10,21]. The development of a biofilm consists of the following sequential events: reversible attachment of planktonic microorganisms to a surface, secondary, irreversible attachment, development of microcolonies by the previously adhered microorganisms, the secretion of the extracellular polymeric substrate, the development of a three-dimensional biofilm community, and finally, the detachment of microorganisms from the biofilm community and dissemination into the environment [22]. Some of the biofilm forming microorganisms that could constitute a source of healthcare-associated infections are: Gram-positive pathogens, such as *Staphylococcus aureus*, mostly found on skin and mucous membranes, causing multiple infections including bacteraemia, infective endocarditis, skin and soft tissue infections, osteomyelitis, septic arthritis, prosthetic device infections, pulmonary infections, gastroenteritis, meningitis, toxic shock syndrome, and urinary tract infections [23,24]; *Enterococcus faecalis* which causes similar infections as the *Staphylococcus aureus* pathogen [25]; and Gram-negative pathogens, such as *Pseudomonas aeruginosa*, a rod-shaped bacterium, causing several diseases in the human body, some of which are bacteraemia, ventilator-associated pneumonia, urinary tract infections and skin soft tissue infections [26], and *Klebsiella pneumoniae*, which is mostly found in the human intestines. It is among the world’s most common nosocomial pathogens, causing pneumonia, UTIs, bloodstream infections, and sepsis [27]. 

An important step in understanding the influence of the dielectrophoretic forces on the formation and growth of the bacterial biofilms within the microfluidic bioreactors is the characterization of the obtained biofilms through quantitative and qualitative means. Some characterization techniques applicable to biofilms include, but are not limited to, quantitative analysis, as well as qualitative analysis using Optical Microscopy (OM), Optical Coherence Tomography (OCT), and Scanning Electron Microscopy (SEM). For the OCT technique, a higher signal intensity is yielded by objects with higher light scattering, therefore allowing the visualization of different shades of black and white on the interface of media with different refractive indexes. By identifying these shades, the existence of the biofilm can be determined [28]. Using a dedicated ThorImage software (ver. 5.5.1. from THORLABS GmbH, Luebeck, Germany) for image processing, comprehensive bidimensional scanning images of the biofilm thickness can be represented. Similarly, the SEM images enable the investigation of the morphology and topography of the biofilm surface [10].

This paper outlines a new approach to studying and influencing the preferential formation and growth of bacterial biofilms in microfluidic bioreactors using dielectrophoresis as means for the manipulation of cells. The investigation focuses on the capacity of four different types of bacteria: *Staphylococcus aureus* ATCC 25923, *Pseudomonas aeruginosa* ATCC 27853, *Enterococcus faecalis* ATCC 29212, and *Klebsiella pneumoniae* ATCC 13883 to form biofilm under the influence of dielectrophoretic forces, inside 3D printed, PLA microfluidic bioreactors.

## 2. Materials and Methods

### 2.1. Device Fabrication and Setup 

A Crealty3D Ender 5 printer (Shenzhen Creality 3D Technology Co., Ltd., Shenzhen, China) was used for manufacturing the microfluidic devices that serve as bioreactors. The design of these devices was created using the SolidWorks 3D CAD Software (Education Edition 2019–2020, Dassault Systèmes, Vélizy-Villacoublay, France) and included an H type channel system, with 2 inlet and 2 outlet holes for the admission and evacuation of the fluids. The system was built to accommodate a copper electrode, Ø 1 mm, parallel to the main channel and 9 copper electrodes, Ø 0.7 mm, perpendicular to it, to create a dielectrophoretic effect inside the system. The electrodes were purchased in the form of copper wires, which were then manually inserted in the specifically designed channels. The design characteristics for the main channel were 38 mm length, 1.5 mm width, and 1.6 mm height (Figure 1 and Figure 2). 

The bioreactors (Figure 2a,b) were printed using a 1.75 mm high performance PLA filament (VerbatimTM—Mitsubishi Kagaku Media Co., Ltd., Tokyo, Japan), heated up to 205 °C, and deposited at a speed rate of 60 mm/s. The devices were fabricated in one single print, using the “bridging” technique, which allows the printer to build “bridges” between structures less than 5 mm tall, by stretching the hot material for short distances and therefore printing with minimal sagging. Consequently, there was no need for support structures in the printing of the central channel, inlet/outlet, or lateral channels. To ensure the complete sealing of the system, inlet/outlet adapters were UV soldered to the system using the Buildfix Pro DC hybrid composite bonding agent (Figure 2c). A Power Supply Bench UNI-T UTP-3703, at an output voltage of 30 V was used to generate the non-uniform electric field. 

The formation and growth of bacterial biofilms was studied using suspensions of *Staphylococcus aureus* ATCC 25923 (SA), *Pseudomonas aeruginosa* ATCC 27853 (PA), *Enterococcus faecalis* ATCC 29212 (EF), and *Klebsiella pneumoniae* ATCC 13883 (KP), together with Thermo Scientific™ (Thermo Fisher Diagnostics SAS, Dardilly, France), Nutrient Broth (Dehydrated), CM0001B nutrient broth. 

To determine the influence of the dielectrophoresis on the formation and growth of the bacterial biofilms, two types of samples were prepared: standard samples, in which the bacterial addition was carried out on the side accommodating the perpendicular electrodes (perpendicular-electrode side (PPE)), and switched samples, where the bacterial suspension was introduced on the parallel-electrode side (PE) of the channel. 

### 2.2. Biofilm Formation and Growth

Prior to the preparation of the bacterial suspensions, the SA ATCC 25923, PA ATCC 27853, EF ATCC 29212, and KP ATCC 13883 strains were cultivated on 5% sheep-blood agar plates and incubated for 24 h at 37 °C. The final solutions were obtained by mixing the colonies picked up from the agar plates in a 10 mL saline solution, at a 1 McF turbidity. 

Volumes of 3 mL of nutrient broth and 3 mL of bacterial suspension were drawn in two syringes, which were then connected to silicon tubes attached to the inlets of the system. Using an SP230iwZ Syringe Pump (WPI), the fluids were pushed through the microfluidic devices at a flow rate of 0.1 mL/min, at room temperature (Figure 3). Approximated parameters for the flow were Reynold’s number of 1.49, flow velocity of 0.69 mm/s and pressure of 2.48 mBar. After the system was started, a 30 V potential (DC) was applied to the electrodes (value based on pervious experimental work presented at RoMAT 2020 conference). This potential was applied until the flow of fluids was stopped within the bioreactor (approximately 30 min). The microfluidic bioreactors were then unplugged from the syringe pump, the silicon tubes were removed, and the microsystems were incubated for 24 h at 37 °C. The final step was the inactivation of the bioreactors using UV light. In preparation for the optical analysis, the samples were cut along the main channel using a Robotec laser cutting machine. 

### 2.3. Analysis Methods

While the standard procedure for quantitative analysis of the biofilms is a microtiter plate technique, in this case an adapted version of it was used. For this, the bioreactors were washed out with 5 mL of saline solution to remove any leftover planktonic cells, and then, a 5 mL 1% crystal violet solution was pushed through the main channel to stain the biofilm. The next steps included a 15 min incubation period at room temperature, 3 consecutive washes with saline solution for the removal of the excess dye, and a final wash with 5 mL of ethanol with acetone to solubilize the stained biofilm. This final solution was collected, diluted 1:10 with ethanol-acetone and assessed at 590 nm using a UV–VIS spectrometer.

To determine the formation of the biofilm inside the microfluidic bioreactors, spectral domain optical coherence tomography (SD-OCT) technique measurements were used. The SD-OCT measurements were recorded using an OCTH-1300 Handheld Scanner (THORLABS GmbH, Luebeck, Germany) working in the 1200–1400 nm domain with the central wavelength of 1300 nm, equipped with an OCTH-LK30 lens kit (THORLABS GmbH, Luebeck, Germany). The lateral and axial resolutions of the SD-OCTH-1300 were 24 µm and 5.5 µm, respectively; the working distance used was 22 mm, and the medium at the interface with the channel surface of the biofilm was air. The scanning speed of the SD-OCTH-1300 was 76 kHz, at a selected refractive index of 1.40 mm, and a pixel resolution of 1.85 μm/px. In addition, an inverted configuration Olympus IX71 (Olympus Corporation, Tokyo, Japan) microscope was used in bright-field (BF) and differential interference contrast (DIC) to visualize in reflectivity the two sides, denoted PPE, and PE, of the channels. The C-mount of the CCD colour camera was 0.63×, and an additional 1.6× magnification lens was used in certain measurements to highlight the details more accurately. For each type of sample with bacterial addition on PE or PPE, 5 microfluidic bioreactors were tested, and within these samples, 5 different positions in the channel were considered for SD-OCT measurements.

Using a JEOL/JSM 5600-LV (JEOL Ltd., Tokyo, Japan) scanning electron microscope (SEM), the topography of the biofilm surface was studied. The images were recorded in Secondary Electron Imaging (SEI) signal at an accelerating voltage of 10 kV. In preparation, the same samples firstly used for OM and SD-OCT analysis were then washed with 5 mL of absolute ethanol and dried at room temperature. The next step was the deposition of a carbon layer on the surface of the biofilm, using a plasma sputtering equipment. The open access ImageJ software (Open-source software, version 1.8.0_172) was used to analyse all recorded OM, SD-OCT, and SEM images.

## 3. Results and Discussions

### 3.1. Distribution of the Electric Field

The electric field distribution inside de microfluidic bioreactor was simulated using the commercial software COMSOL Multiphysics v.4.3 (version 4.3, COMSOL Inc., Stockholm, Sweden). The parallel electrode was set at ground potential, and the perpendicular electrodes were set at 30 V. All the other surfaces were set to insulating boundary conditions. Figure 4a shows the electric field distribution (V/M), and Figure 4b shows the distribution of the electrical potential (V) inside the bioreactor. As it can be observed, the purpose of the copper electrodes displayed perpendicular to the main channel was to force the concentration of the electric field in points, as compared to the PE side, where the distribution of the electric field was continuous along the channel (Figure 4c). Figure 4a,d show that the higher gradient regions were located at the first and last electrodes, as well as at the tip of the other perpendicular electrodes. The discontinuous distribution of the perpendicular electrodes ensures a non-uniform electric field, and consequently, the cells traversing the central channel are subjected to dielectrophoretic forces. 

### 3.2. Optical Microscopy

The formation and attachment of biofilms inside the microfluidic bioreactors were investigated using OM images, SEM images, SD-OCT images, and the ImageJ software. Figure 5c shows the optical microscopy images of the original channel (Figure 5a), as well as the PPE (Figure 5b) and PE (Figure 5c) sides of the main channel, for the SA sample. All three images were recorded in bright field, in reflectivity, using a 4× objective at a magnification of 2.52×, in the portion located immediately at the entrance to the main channel. These images offer an overview on the scale of evolution of the biofilm within the two sides of the main channel by comparison with the original channel. The yellow bright rectangle visible on the right side of all three images represents the inlet channel. As illustrated by Figure 5b, on the PE side of the channel the filament deposition lines are clearly visible, with a size of around 97.86 μm, which is very close to that of 100 μm of the filaments of the original channel, indicating either no deposition or the deposition of a very thin biofilm layer. In comparison, Figure 5c shows the existence of small, irregular formations that cover the deposition lines, suggesting the attachment and development of a thicker biofilm. 

Figure 6 shows detailed optical microscopy images recorded inside of the biofilm using the additional 1.6× magnification lens for the objectives 4×, in bright field (Figure 6a), and 20× in DIC (Figure 6b), respectively. As illustrated in Figure 6a at 4.032× magnification, the development of the film is uneven along the channel, both in terms of topography and thickness. These aspects are revealed by the variations of the image clarity in the focal plane. Upon closer observation, in the optical image of the biofilm recorded at 20.16× magnification and differential interference contrast, clearly defined formations with variable sizes from 1 μm to 60 μm can be identified (Figure 6b). The large distribution of the sizes of formations inside of the biofilm indicates the presence of both individual cells and their agglomeration in the form of colonies.

### 3.3. Spectral Domain Optical Coherence Tomography

SD-OCT images were used to determine the evolution of the biofilm thickness inside the main channel for the SA samples. Figure 7a–c show the SD-OCT images of the PPE side of the main channel, taken at different distances from the inlet channels. Similarly, Figure 7d–f show the SD-OCT images taken within the PE side of the main channel. These images show the formation of a biofilm layer on top of the PLA filament. Using the ImageJ software, the thickness of the biofilm, as shown in the SD-OCT images, was measured. The results were plotted against the results from the opposite channel (Figure 8) to show the evolution of the biofilm across the same channel at different distances from the inlet, and to compare the differences in the thickness of the biofilm from the two sides. It can be observed that on both sides, the biofilm thickness varies along the channel, with a thicker consistency in the middle of the channel. As presented by the four graphs in Figure 8, the thickness of the biofilm on the PPE side of the main channel is greater than on the PE side. This is an indicative of the dielectrophoretic forces acting on the cells and attracting/pushing them towards the high field gradient region and therefore suggesting a positive dielectrophoresis. 

### 3.4. Scanning Electron Microscopy 

SEM images enable the comparison between the morphology and topography of the biofilms formed inside the PPE and PE sides of the microfluidic channels, as well as a standard static growth on PLA plates. Figure 9a–d, Figure 10a–d, Figure 11a–d, and Figure 12a–d show the PPE side of the standard and switched samples at magnifications of ×100 and ×1000, while Figure 9e–h, Figure 10e–h, Figure 11e–h, and Figure 12e–h show the PE side of the same channels at simialr magnifications, for all the bacterial strains (Figure 9 SA, Figure 10 PA, Figure 11 EF, and Figure 12 KP). 

For the SA biofilms, it can be observed that for the standard sample more biofilm was formed on the PPE side of the channel (Figure 9a,b), whereas for the switched sample (Figure 9c,d), on the PPE side the filament deposition lines are observable, indicating less biofilm attached to the surface. This could be explained by the position of the analysed area, situated directly at the entrance of the main channel. This area was chosen to gain more understanding on the transition between the addition region and the region where the biofilm starts to attach to the surface. For the PA, EF, and KP samples, the analysed area was moved more towards the central region of the channel, therefore allowing for a better visualization of the biofilm. For the EF and KP biofilms regardless of the side chosen for the admission of the bacterial suspensions, the cells responded to the influence of the non-uniform electric field, being attracted to, and pushed towards the side accommodating the perpendicular electrodes (towards the high field gradient region). A higher coverage of the biofilm across the main channel can be noticed for both EF and KP, with almost fully coated channels on the PPE side (Figure 11a,b, and Figure 12a,b) and more visible deposition lines on the PE side for the standard samples. Similarly, for the switched samples, the PPE side displays a higher biofilm coverage of the channel, whereas the PE side depicts less biofilm formed inside the channels. A similar situation can be observed for the PA samples but with a less defined differentiation between the two sides. 

Changes in the biofilm structure and morphology due to environmental factors such as flow rate and shear stress were observed and compared using samples grown under static conditions. Cells grown under static conditions were not subjected to dielectrophoresis. Figure 13 presents the SEM images of the SA, PA, EF, and KP biofilms grown on PLA plates at magnifications of ×1000 (Figure 13a,c,e,g) and ×5000 (Figure 13b,d,f,h). The biofilms grown on PLA plates were more defined, with identifiable cell clusters, comparable with the ones presented in the literature. The shapes of the bacterial cells are well observable, together with the extracellular matrix, as well as some areas with dried salts from the dilution media. In comparison, the biofilms formed inside the microfluidic devices seem less defined, with cells incapsulated in the matrix. This could be explained by the lower shear stress in the proximity of the walls, which facilitates the attachment of the cells to the surface, and the flow velocity which likely washes away the cells that could not get attached to the walls, leaving the remaining ones enveloped in the matrix as a protective step against the forces acting upon them. Another issue worth mentioning is the investigation technique itself. Due to the preparation processes the samples undergo to be investigated by SEM, they go through several stages of different intensities of vacuum, which could lead to the distortion of the cells, and therefore, they no longer appear to have the original shape and size in the images.

### 3.5. Quantitative Analysis

The graph below (Figure 14) shows the concentration of biofilms in the PE side of the channels determined using the adjusted microtiter plate technique. The results represent an average of measurements carried out on 5 channels for each type of sample, standard and switched, for each bacteria type: SA, PA, EF, and KP. As it can be noticed, for each of the bacterial strains, switching the admission inlet for the bacterial suspension leads to a drop in biofilm concentration, indicating that the cells are repelled by the electric forces induced by the parallel electrode and therefore suggesting a positive dielectrophoretic answer of the bacterial cells. 

## 4. Conclusions

*Staphylococcus aureus*, *Pseudomonas aeruginosa*, *Enterococcus faecalis*, and *Klebsiella pneumoniae* biofilms have been successfully grown in 3D printed PLA microfluidic bioreactors under the influence of dielectrophoretic forces. The influence of a non-uniform electric field upon the formation of biofilms has been observed using qualitative and quantitative techniques. The COMSOL simulations render the concentration of the electric field in points on the PPE side, as opposed to the uniform distribution of the electric field on the PE side indicating the presence of a non-uniform electric field and therefore of the dielectrophoretic forces inside the microfluidic channel. Based on the images and graphical representation of the biofilm concentration, we can conclude that all the bacterial cells display a positive dielectrophoretic behaviour, with a thicker biofilm present in the channel where the electrodes were displayed perpendicular to the main channel (in high field gradient region). 

Biofilm structure and topography is noticeably different for the biofilms grown under dynamic conditions, compared to those grown on plates, phenomena explained by the shear forces and flow velocity that act upon the cells. 

The present findings have important implications for future research implying the study of formation and growth of biofilms in in situ conditions. Comprehending the systems that allow the development and dissemination of biofilms could lead to understanding their behaviour in different environments and therefore to improvements in the fighting mechanisms against foul biofilms. Moreover, by using the same experimental setting, optimization of the antibiotics treatment against bacteria and fungi can be achieved.

## Figures and Tables

**Figure 1 micromachines-13-01377-f001:**
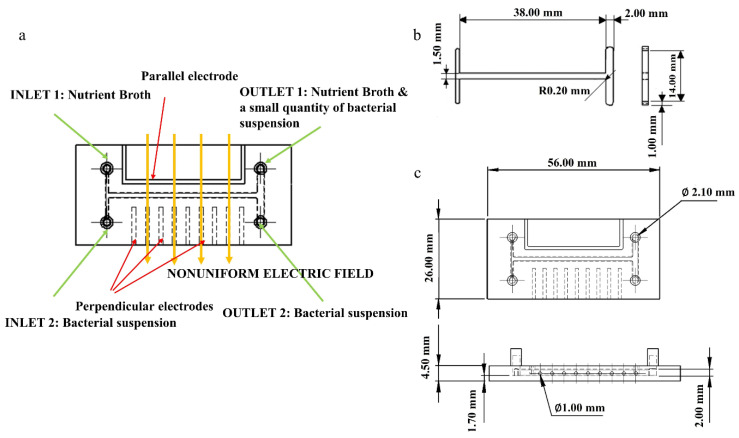
Schematic representation of the microfluidic bioreactor: (**a**) overview of the main components of the device; (**b**) overall dimensions of the central channel; (**c**) top and side view of the microfluidic bioreactor, with overall dimensions.

**Figure 2 micromachines-13-01377-f002:**
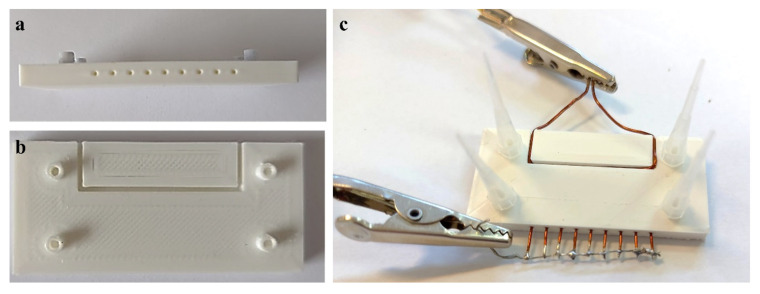
Printed microfluidic bioreactor: (**a**) side view; (**b**) top view; (**c**) microfluidic bioreactor with integrated electrodes.

**Figure 3 micromachines-13-01377-f003:**
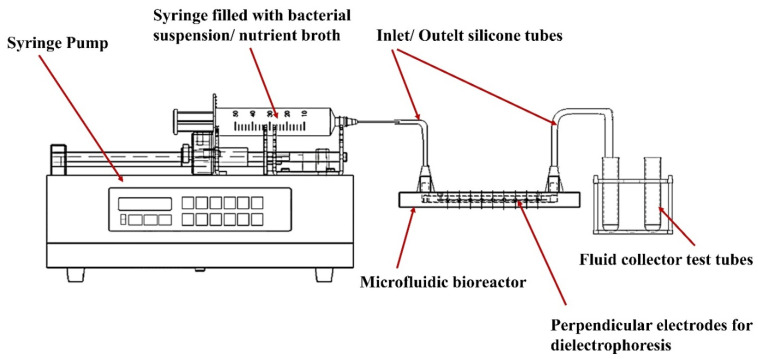
Schematic representation of the device setup, including a syringe pump, the syringes filled with bacterial suspension and nutrient broth, the microfluidic bioreactor, inlet/outlet silicone tubes, perpendicular electrodes, and the test tubes for final fluid collection.

**Figure 4 micromachines-13-01377-f004:**
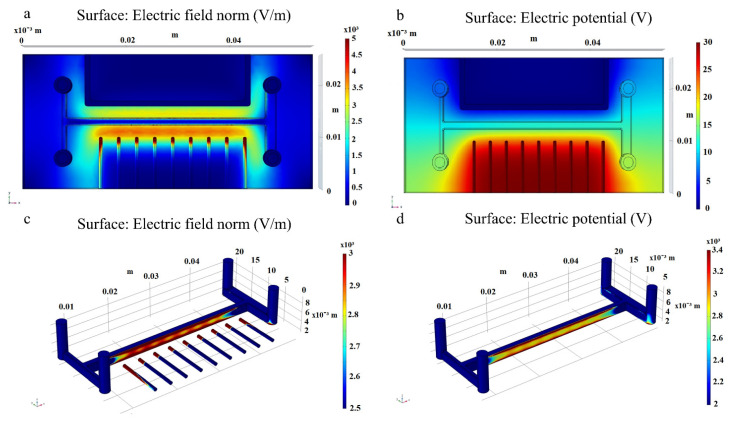
Electric field distribution (V/m) inside the bioreactor: (**a**) top view; (**c**) central channel-PE side; (**d**) central channel–PPE side, and (**b**) distribution of the electrical potential (V) within the microfluidic bioreactor (top view).

**Figure 5 micromachines-13-01377-f005:**
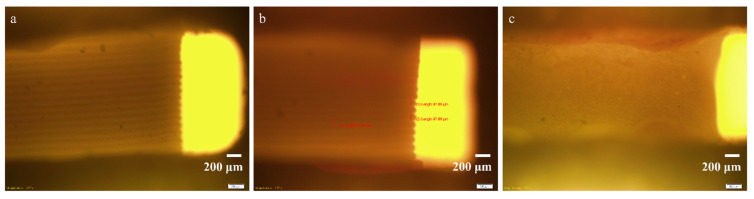
Optical images recorded in bright field, in reflectivity at a magnification of 2.52× for the original channel (**a**); PE side (**b**) and PPE side of the channel (**c**).

**Figure 6 micromachines-13-01377-f006:**
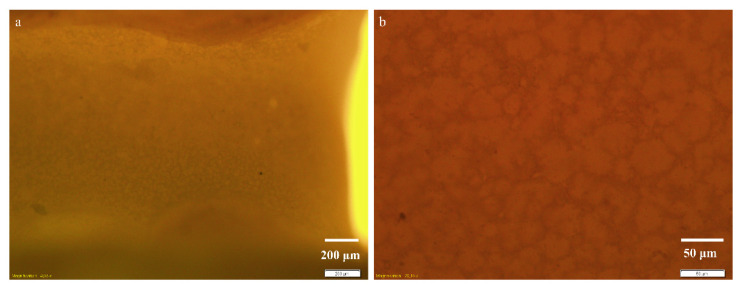
Optical images recorded in biofilm, on the PPE side of the main channel, at increasing magnifications of (**a**) 4.032×, in bright-field, and (**b**) 20.16×, in DIC, respectively.

**Figure 7 micromachines-13-01377-f007:**
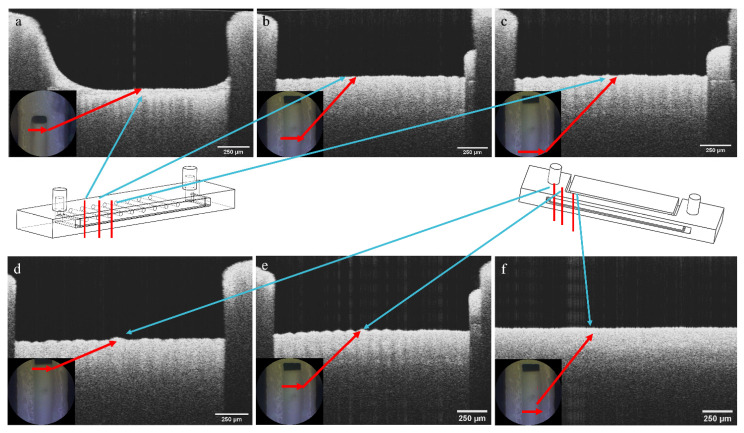
SD-OCT images of the (**a**–**c**) PPE side of the main channel; (**d**–**f**) PE side of the main channel of the microfluidic device, with marked distances at which the measuremets were carried out.

**Figure 8 micromachines-13-01377-f008:**
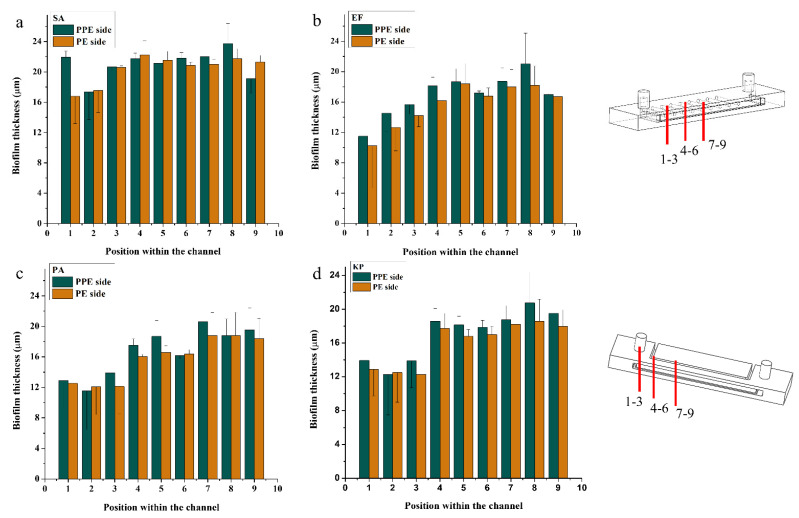
The evolution of the biofilm thickness within PPE and PE sides of the channels at different distances from the inlet (1–3 inlet area, 4–6 middle of the channel, 7–9 end side of the channel) for: (**a**) SA, (**b**) EF, (**c**) PA, and (**d**) KP biofilms.

**Figure 9 micromachines-13-01377-f009:**
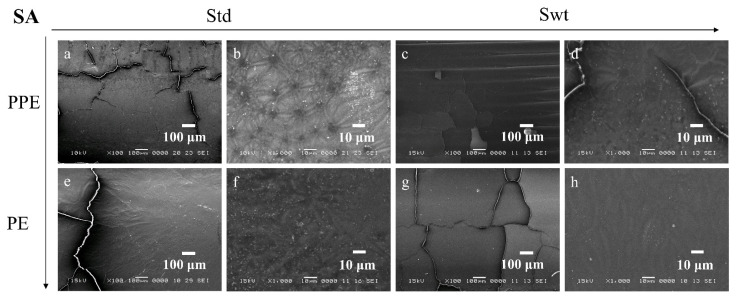
SEM images of SA biofilms in standard (**a**,**b**,**e**,**f**) and switched (**c**,**d**,**g**,**h**) samples on (**a**) ×100, (**b**) ×1000, (**c**) ×100, and (**d**) ×1000 PPE side; (**e**) ×100, (**f**) ×1000, (**g**) ×100, and (**h**) ×1000 PE side of the channel.

**Figure 10 micromachines-13-01377-f010:**
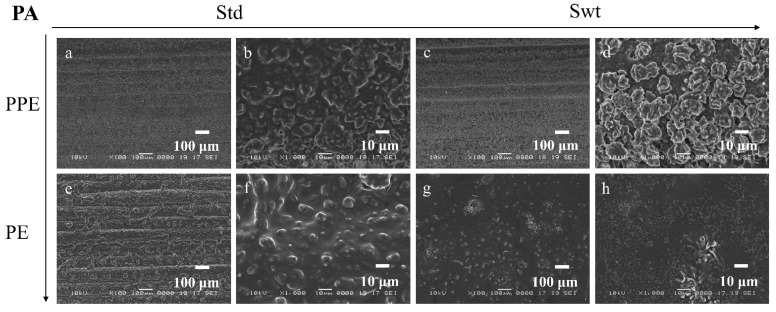
SEM images of PA biofilms in standard (**a**,**b**,**e**,**f**) and switched (**c**,**d**,**g**,**h**) samples on (**a**) ×100, (**b**) ×1000, (**c**) ×100, and (**d**) ×1000 PPE side; (**e**) ×100, (**f**) ×1000, (**g**) ×100, and (**h**) ×1000 PE side of the channel.

**Figure 11 micromachines-13-01377-f011:**
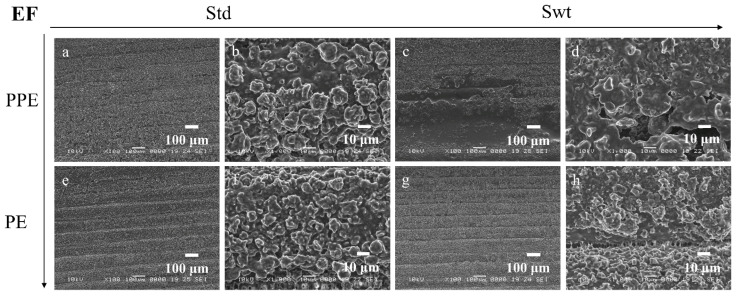
SEM images of EF biofilms in standard (**a**,**b**,**e**,**f**) and switched (**c**,**d**,**g**,**h**) samples on (**a**) ×100, (**b**) ×1000, (**c**) ×100 and (**d**) ×1000 PPE side; (**e**) ×100, (**f**) ×1000, (**g**) ×100, and (**h**) ×1000 PE side of the channel.

**Figure 12 micromachines-13-01377-f012:**
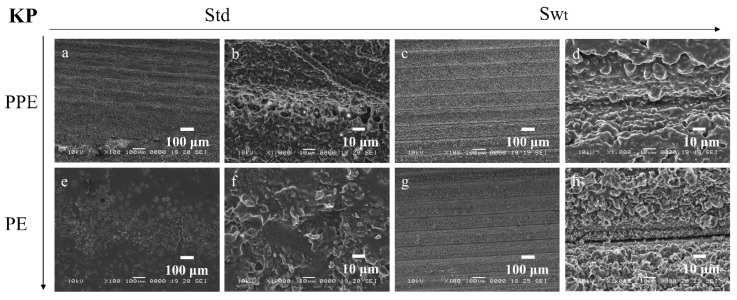
SEM images of KP biofilms in standard (**a**,**b**,**e**,**f**) and switched (**c**,**d**,**g**,**h**) samples on (**a**) ×100, (**b**) ×1000, (**c**) ×100 and (**d**) ×1000 PPE side; (**e**) ×100, (**f**) ×1000, (**g**) ×100, and (**h**) ×1000 PE side of the channel.

**Figure 13 micromachines-13-01377-f013:**
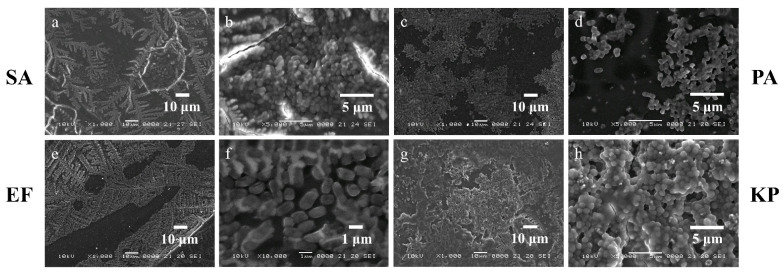
SEM images of biofilms grown under static conditions for SA (**a**,**b**), PA (**c**,**d**), EF (**e**,**f**), and KP (**g**,**h**) at magnifications of ×1000 (**a**,**c**,**e**,**g**), and ×5000 (**b**,**d**,**f**,**h**).

**Figure 14 micromachines-13-01377-f014:**
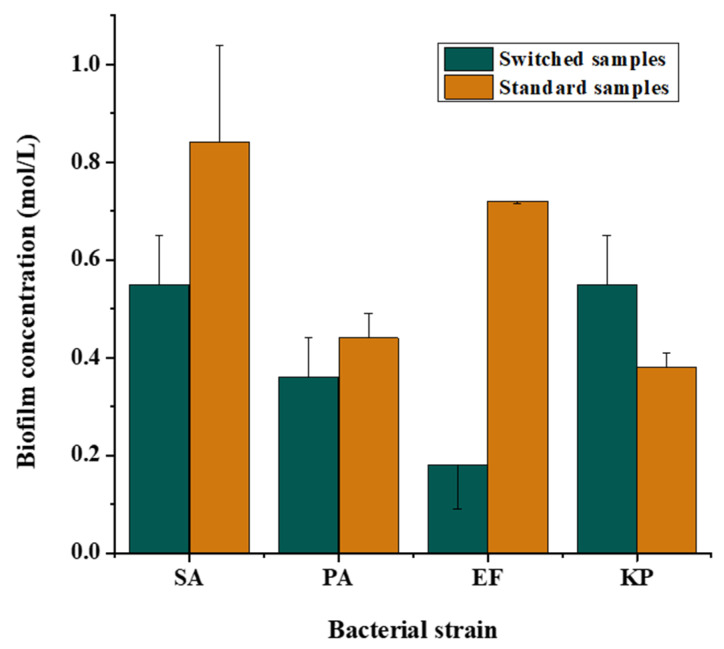
Concentration of the biofilms within the PE side of the channel for each bacterial strain, in standards and switched samples.

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
