# Peer review of "3D Printed Microfluidic Bioreactors Used for the Preferential Growth of Bacterial Biofilms through Dielectrophoresis"

_micromachines, 2022, doi:10.3390/mi13091377_

Round 1

Reviewer 1 Report

Please see the profile.

Author Response

Thank you for giving us the opportunity to submit a revised draft of the manuscript: “3D printed microfluidic bioreactors used for the preferential growth of bacterial biofilms through dielectrophoresis” for publication in the Journal Micromachines. We appreciate the time and effort that you dedicated to providing feedback on our manuscript and are grateful for the insightful comments on and valuable improvements to our paper. We have incorporated most of the suggestions made by the reviewer. Those changes are highlighted within the manuscript.

Reviewer 2 Report

The paper presents a 3D printed microfluidic device that is used to preferentially grow bacterial films. The device also uses DEP for the process. While the idea is interesting, the paper is not written very clearly. Fundamental things like schematic diagrams of the device and electrode geometry are missing. It is rather difficult to understand the actual structure of the device from the text. It is also not mentioned how the electrodes were integrated within the 3D printed device. No photograph of the device is presented. While the work seems to have technical merit, I cannot recommend it for publication due to the presentation quality of the manuscript. I recommend the authors to resubmit the paper after proper modifications. Some of my detailed comments are listed below:

1.     In line 83-85, it is stated that particle moves toward field maxima for positive DEP and particle moves toward field minima for negative DEP. This is incorrect. For positive DEP, a particle moves toward high field gradient region (which is unrelated to field maxima). For negative DEP, a particle moves away from high field gradient region (which is unrelated to field minimum). Please correct this statement in the manuscript.

2.     The image quality of Fig. 2 is not satisfactory. The image appears blurry and out of focus. Please replace the figures with higher quality images. Also, label the features on the figure.

3.     A schematic diagram of the device should be included. It is difficult to understand the device structure without a diagram.

4.     Actual photos of the 3D printed device should be included in the manuscript.

5.     How were the electrodes integrated within the 3D printed device? What are the dimensions of the electrodes?

6.     The y axis in Fig. 5 is illegible.

7.     FDM printed PLA based microfluidics sometimes suffer as the surface is not smooth. This can result in lack of seal within microfluidic channels. Resin based printers (SLA) often perform better. Please discuss why FDM 3D printing method was selected for this work.

8.     The authors may choose to cite a few more papers on dielectrophoresis. That will give a reader further information about DEP. A few papers on the topics are:

a.     https://doi.org/10.1021/ac070810u

b.     https://doi.org/10.1063/5.0049126

Author Response

(The authors gave the same response as above.)

Reviewer 3 Report

This paper use a 3D printed microfluidic device for studies of biofilm formation under flow in combination with DEP. They attempt to evaluate the thickness of the biofilms for different bacteria species and if the applied voltage affects the thickness.

Unfortunately, the paper does not reach the standards for a scientific publication. The text must be improved: the introduction fails to present the significance and novelty of the work. The materials and methods lacks structure and important information on e.g. the device design and experimental procedure of DEP. The results are presented for, as far as I understand, only one experiment per bacteria species, which is too little to draw any conclusions on the differences in biofilm thickness, although these preliminary data indicate that there is a difference in thickness depending on side.

Some specific comments:

Many typos, e.g.  56 designed, 67 improve, 86 therefore, 92 substrate

No need for fig 1.

On line 91: This is one way, in vitro. In my opinion, detachment is not part of formation. Are you detaching your biofilms?

On line 114: This is not specific to biofilms

The information that is now in the introduction can be more concise and some of it is not needed.

Sometimes the paragraphs feels a bit randomly placed. This would be improved if you add topic sentences. For example on line 98

I would suggest rewriting the introduction according to CARS, so that the reader gets an overview of the significance and novelty of the research presented in the article. It is perhaps not necessary to go through all methods and material here, could be just mentioned more briefly. Also, add references to relevant previous work.

The introduction must answer why you want to manipulate biofilms with DEP.

Add a schematic of the device with fluidic channels and electrodes clearly illustrated. How are the electrode fabricated and what material?

What is the electrical field in the device based on your 30 V? I suggest to add a COMSOL simulation, or similar, showing the electrical field. How was the electrical field applied? When and for how long? Did you try different voltages? Is is AC or DC? If AC what is the frequency?

Please, do not refer to the left and right side, find a more scientific nomenclature e.g. point-electrode-side and parallel-electrode-side. Or low and high (electrical) field side

Section 2.1 3D printing: this contains too much information not related to 3D printing, please restructure. It can be called e.g. device fabrication or setup, but then move information on bacteria.

Did you apply any flow through the chip during the 24 h that the chip was incubated?

Fig 2. I can’t make sense of this figure. Why is it blurry? What is in focus? From what angle are we looking? What are the features, including the orange, darker, yellow blub? For reference, add also same image without biofilm.

In figure 3, add a schematic showing the positions of a, b, c, d, e, f. Make the text of the scale bar larger so that it is easier to read

Figure 4: did you only measure on one chip? There is a lot of noise in the data. Do you have significance in your statement that the biofilm is thicker on the one side? I am not convinced. I would recommend repeated measurements and plotting the average and standard deviation for each point. Also ad unit on the x-axis

Figure 5 plots exactly the same data as figure 4 and is redundant. You can add a grid in fig 4 if you want it to be easier to read and compare.

Was dielectrophoresis applied on the static control? Also, did you make a microfluidic control without electrical field?

Is figure 11 made with one sample for each column? It would gain a lot of having repeats of the experiment.

How did you quantify that the bacteria grew inside the chip? Was there more in the films than what was inserted?

Present the data of the thicknesses of the different biofilms, maybe in a table. Explain why you think you see this difference in more detail.

Have the authors considered if the applied voltages cause electrolysis on the electrodes and gas formation? This has been used as a method for cleaning sensor electrodes from biofilms. Please verify that this does not happen in your setup.

Author Response

(The authors gave the same response as above.)

Reviewer 4 Report

The manuscript titled “3D printed microfluidic bioreactors used for the preferential growth of bacterial biofilms through dielectrophoresis.”  claims to have investigated the influence of the dielectrophoresis on the formation and growth of Staphylococcus aureus ATCC 25923, Enterococcus faecalis ATCC 29212, Pseudomonas aeruginosa ATCC 27853, and Klebsiella pneumoniae ATCC 13883 biofilms inside 3D printed microfluidic devices.

The work lacks scientific novelty in microfluidics, bioreactor and biofilm sectors simultaneously. The manuscript is poorly written as well as the insufficiency of explanation; scientific data makes the manuscript misconceive. Therefore, I recommend the rejection of this article.

More specifically:

1)   The abstract of an article should be the basic overview of the whole work. Line 13-16 has a citation and theme that fits the needs of the Introduction of this article.

2)   The title is self-explanatory. The fabrication/model/diagram or anything related to microfluidics is nonexistent.

3)   The author claims to have developed a microfluidic device which works Bioreactor 131-132, but there is no expiation. Moreover, the term Bioreactor was only used 2 times!

4)    Then again, lines 49-52 statement conflict with the following statements 54-62 (The overall process explained it is quite expensive. The author should provide the developed device’s cost and compare with their review citation 13’s Although in 49-52 which part of the review the authors are talking about is confusing.

5)   The context of Figure 1 is unclear. If they used the method of M. Murariu and P. Dubois from citation number 15, The authors should use their own schematic.

6)   Due to the lack of expiation of the device fabrication and simulation, it is hard to understand the figures in this article. Authors should consider the structure of the paper and explain it further.

7)   The authors used past tense in lines 193-194, but 216-219 were written in the present tense. There are many other places the tenses are not used properly. This manuscript needs a major English revision.

8)   In the conclusion line, 333-336 authors claimed thicker dielectrophoretic behaviour from the bacteria, but their data is unclear and not explained. A detailed cross-section and integrated figure with a graph might be helpful. Better use the name of the cross-section in the chart.

9)   An assembled microfluidic device as a figure (Maybe a Schematic) or a graphical image of the over study is needed to understand the work further.

10)  A grammar check is needed. For instance lines 23-25, the verb ‘was’ is used, which does not agree with the statement of a plural form.

11)  Typo’s need further attention Typo: therefor (line 86)

Author Response

(The authors gave the same response as above.)

Round 2

Reviewer 2 Report

The authors have significantly improved the paper. However, there are still some issues that should be addressed. My detailed comments are listed below:

1.     The quality of Fig. 1 needs to be significantly improved. The resolution is low and the text can be difficult to read. Also, the units of the dimensions were not mentioned. I am assuming that the units are mm. If it is not possible to insert the units in the figure, it should at least be mentioned in the figure caption. Also, the different images in Fig. 1 should be labelled as (a), (b).. etc. and the caption should state the different views.

2.     Some cross-sectional dimensions are missing in Fig. 1. Please note the side view (Fig. 1, bottom right). It is not stated what is the height of the liquid chamber, the height of the top surface or the thickness of the bottom surface. Only the combined height of 4.5 mm is stated. This is not sufficient as the aforementioned key information are missing.

3.     It appears that the entire microfluidic device was fabricated by a single 3D print. I initially assumed that the top side and the bottom side of the device were printed separately and then bonded/glued together. It should be clearly stated in the manuscript that the entire device was fabricated by a single 3D print. This, however, introduces new questions. 3D printing often produces support structures in the cavities that have to be later removed manually. One would assume that there would be support structures within the fluid chamber and the microfluidic channels. Since the final 3D printed device is sealed, how is it possible to remove those support structure from inside the device? This seems like a major issue. Please discuss this in detail.

4.     The author states that the perpendicular electrodes have a diameter of 0.7mm and they were inserted into the 3D printed device. Although this information was in the author response, the manuscript omits the part about the insertion. Was it manually inserted? Please discuss briefly in the manuscript. Also, please mention if the electrodes were purchased or manufactured (and more details about it).

5.     Due to the device structure, it appears that the images in Fig. 5 were taken through the top PLA surface. Hence it makes sense that the figure quality is not very good. Please mention the thickness of the top PLA surface through which the image was taken.

6.     Why wasn’t a transparent filament used to 3D print the device? That would have made the imaging much simpler. Please explain.

7.     Was the SEM performed when the biofilm was inside the device? I would assume that to be very challenging. An SEM imaging through a PLA plastic layer does not seem practical. Was the biofilm taken out of the device to do the SEM imaging. Please explain in detail.

8.     A schematic of the optical imaging setup should be included in the paper.

9.     It is mentioned that 30V was applied between the electrodes. Is this a DC voltage or an AC voltage. In case of AC voltage, please mention the frequency of the voltage. In case of DC voltage, further explanation is needed. A DC voltage may lead to electrolysis. How was this avoided?  

Author Response

(The authors gave the same response as above.)

Author Response

(The authors gave the same response as above.)

Round 3

Reviewer 2 Report

The paper has been improved. It may be considered for publication at the editor's discretion.